# How to More Effectively Obtain Ginsenoside Rg5: Understanding Pathways of Conversion

**DOI:** 10.3390/molecules28217313

**Published:** 2023-10-29

**Authors:** Leqin Cheng, Wei Luo, Anqi Ye, Yuewei Zhang, Ling Li, Haijiao Xie

**Affiliations:** 1Jilin Institute of Chemical Technology, School of Chemistry and Pharmaceutical Engineering, Jilin 132022, China; 2Tonghua Bai’aojinsen Biotechnology Co., Ltd., Tonghua 134000, China; 3Hangzhou Yanqu Information Technology Co., Ltd., Hangzhou 310003, China

**Keywords:** rare ginsenoside Rg5, conversion pathway, DFT calculations, NOESY analysis

## Abstract

Ginsenoside Rg5, a relatively uncommon secondary ginsenoside, exhibits notable pharmacological activity and is commonly hypothesized to originate from the dehydration of Rg3. In this work, we compared different conversion pathways using Rb1, *R*-Rg3 and *S*-Rg3 as the raw material under simple acid catalysis. Interestingly, the results indicate that the conversion follows this reaction activity order Rb1 > *S*-Rg3 > *R*-Rg3, which is contrary to the common understanding of Rg5 obtained from Rg3 by dehydration. Our experimental results have been fully confirmed by theoretical calculations and a NOESY analysis. The DFT analysis reveals that the free energies of S-Rg3 and R-Rg3 in generating carbocation are 7.56 mol/L and 7.57 mol/L, respectively, which are significantly higher than the free energy of 1.81 mol/L when Rb1 generates the same carbocation. This finding aligns with experimental evidence suggesting that Rb1 is more prone to generating Rg5 than Rg3. The findings from the nuclear magnetic resonance (NMR) analysis suggest that the fatty chains (C22–C27) in R-Rg3 and S-Rg3 adopt a Gauche conformation and an anti conformation with C16–C17 and C13–C17, respectively, due to the relatively weak repulsive van der Waals force. Therefore, the configuration of R-Rg3 is more conducive to the formation of intramolecular hydrogen bonds between 20C–OH and 12C–OH, whereas S-Rg3 lacks this capability. Consequently, this also explains the fact that S-Rg3 is more prone to dehydration to generate Rg5 than R-Rg3. Additionally, our research reveals that the synthetic route of Rg5 derived from protopanaxadiol (PPD)-type ginsenosides (including Rb1, Rb2, Rb3, Rc and Rd) exhibits notable advantages in terms of efficacy, purity and yield when compared to the pathway originating from Rg3. Moreover, this study presents a highly effective and practical approach for the extensive synthesis of Rg5, thereby facilitating the exploration of its pharmacological properties and potential application in drug discovery.

## 1. Introduction

Ginsenosides are important representative components of *Panax* plants, and they have extensive pharmacological effects [1,2,3,4,5]. Specifically, certain secondary rare ginsenosides with low polarity, including Rg5, Rg3 and Rh2, exhibit heightened biological efficacy in the realms of anti-cancer, anti-inflammatory, and antidepressant properties as a result of the transformation of ginsenosides [2,6,7]. Therefore, considerable effort has been dedicated to the conversion of a major ginsenoside into a secondary ginsenoside [8,9,10,11,12,13].

Ginsenoside Rg5 (Figure 1) was first isolated from processed ginseng by a scientific research team in South Korea in 1996 [14]. Numerous studies have demonstrated that ginsenoside Rg5 possesses the capability to effectively trigger apoptosis in cancer cells [15,16,17,18], while also exhibiting a wider and more potent medicinal potential in diabetes [6,19], hypertension [20] and anti-inflammatory [1,21]. Furthermore, it is evident that ginsenoside Rg5 exhibits a higher efficacy compared to ginsenoside Rg3 in enhancing memory [22] and anti-cancer activity (including human lung cancer cells (NCI-H460), colorectal cancer cells (CACO-2), hepatocellular carcinoma cells (SMMC-7721), gastric cancer cells (SGC-7901), and breast cancer cells (MCF-7)) [23]. The noteworthy pharmaceutical properties exhibited by Rg5 have garnered substantial interest among professionals in the fields of medicinal chemistry and pharmacology.

Ginsenoside Rg5 is the degradation product of protopanaxadiol (PPD)-type saponins. Based on an analysis of the chemical structure and properties, it is evident that Rg3 has the potential to undergo dehydration at the C20 position, resulting in conversion into Rg5. Therefore, the utilization of Rg3 dehydration as a means to produce Rg5 has gained significant recognition and credibility within the academic community [9,10].

For example, Li’s team [24] synthesized Rg5 from Rg3 utilizing formic acid as a catalyst through a 6 h heating reaction in an ethanol solvent, resulting in a yield of 16.4%. Liu et al. [23] treated the total ginsenoside with cellulase for 30 h to prepare ginsenoside Rg3 and then dehydrated it for 6 h under the catalysis of citric acid to prepare ginsenoside Rg5, with a yield of 27.0%. In other words, it takes a longer time to prepare Rg5 by dehydration of ginsenoside Rg3, and the yield is relatively low. We believe that most experimental data are accurate (Figure 1), but the controversy is associated with how Rg5 is converted. Why is the yield of Rg5 prepared by dehydration of Rg3 very low? Is it necessary to generate Rg5 through the intermediate product Rg3?

R-Rg3 and S-Rg3 (Figure 1) are hydrolysis products of ginsenoside Rb1, and they are also considered to be the main source of Rg5 by many people [9,10,23]. The clarification of the distinct activities exhibited by R-Rg3, S-Rg3 and Rb1 in the production of Rg5 can provide valuable insights into comprehending the precise pathway of Rg5 generation. Therefore, we designed an experiment to prepare ginsenoside Rg5 using ginsenoside Rb1, R-Rg3 and S-Rg3 as raw materials by hydrochloric acid-catalyzed in methanol to investigate the underlying mechanism involved in the formation of ginsenoside Rg5. Meanwhile, density functional theory (DFT) and structural analysis were used to explore the main reasons for the different activities of S-Rg3, R-Rg3, and Rb1 in generating Rg5. These studies can offer theoretical backing for the cost-effective and extensive production of ginsenoside Rg5, while also establishing a material basis for further exploration of its pharmacological properties and the development of pharmaceutical agents.

## 2. Results

### 2.1. Preparation of Ginsenoside Rg5 from Single Ginsenoside S-Rg3, R-Rg3 and Rb1

Ginsenoside S-Rg3 and R-Rg3 were treated with hydrochloric acid in methanol solvent and the reaction products were analyzed by HPLC at 20 min, 40 min and 60 min, respectively (Figure 2A,B). According to the findings depicted in Figure 2A,B and Appendix A, ginsenoside Rg5 were obtained by dehydration reaction of S-Rg3 within 60 min, while only a trace amount of Rg5 was obtained when R-Rg3 was used as raw material. This finding suggests that the utilization of R-Rg3 is not appropriate for the preparation of ginsenoside Rg5.

Our results showed that S-Rg3 could be partially converted into Rg5; however, the conversion rate and yield were 38.97% and 18.81%, respectively. In distinct contrast, when R-Rg3 was used as the raw material, the conversion rate and yield were lower, only 10.59% and 3.04% (Appendix A). Thus, we turned our attention to the raw materials that could also generate products from the reaction mechanism, PPD. Ginsenoside Rb1 is one of the main saponins in PPD saponins; therefore, we first chose that raw material for our research.

Excitingly, most of Rb1 was rapidly converted to Rg5 in just 20 min (Figure 2C and Appendix A). However, only 38.97% of S-Rg3 was converted to a small amount of Rg5 during the same period of time. This finding suggests that the ginsenoside Rb1, compared to Rg3, can be more easily converted to Rg5 with a higher yield. This also indicates that ginsenoside Rb1 may be more suitable for preparing Rg5 as a raw material.

### 2.2. Preparation of Ginsenoside Rg5 from PPD-Type Saponin Mixture

PPD-type saponins in ginseng root saponins mainly include Rb1, Rb2, Rb3, Rc and Rd. It is cheaper than single ginsenoside Rg3 and Rb1. The ginsenosides in PPD saponin have similar structures, both of which have glycosidic bonds at positions 3C and 20C. In order to investigate the activity of PPD-type saponins to generate Rg5, the reactions were carried out using a PPD saponin mixture as a raw material in the presence of an acid catalyst. From Figure 3 and Appendix A, we found that TLC and HPLC analysis of the PPD saponin mixture’s reaction products was the same as for the Rb1 monomer saponin. If methanol was used as solvent, PPD saponin fully transformed in 20 min, and the majority product was ginsenoside Rg5 with less S-Rg3.

### 2.3. Amplification Experiment of Prepared Rg5

In order to further verify the feasibility of preparing Rg5 from PPD saponins, the reaction was carried out under equivalent conditions by amplifying the dosage of raw materials by 200 times. The experimental results showed that the yield of Rg5 reached 47.71%, slightly higher than that of 45.42% before amplification (see Appendix A).

The findings from experimental investigations suggest that the dehydration of Rg3, particularly R-Rg3, as a means of producing Rg5 is unsuitable due to its low yield, inefficiency, lack of operational convenience, and high cost. In contrast, the method of preparing Rg5 using the PPD saponin mixture showed more advantages, including a lower reaction temperature, shorter time and higher yield, making it the most effective, economical and simple method.

## 3. Discussion

### 3.1. Conversion Pathways Proposed for Ginsenoside Rg5

Ginsenoside Rg3 is a hydrolysis product of ginsenoside Rb1, with a tertiary alcohol structure at the C20 position of Rg3. Therefore, based on the habitual thinking that tertiary alcohols are prone to dehydration under acidic conditions, most people believe that Rg5 mainly comes from ginsenoside Rg3, namely Rb1 → Rg3 → Rg5 [9,10,23]. Is Rb1 really converted into Rg5 through the intermediate of Rg3? If ginsenoside Rb1 is first hydrolyzed to Rg3 and then dehydrated to Rg5, according to the understanding of most people, the time required for Rb1 to Rg5 must be longer than Rg3 to Rg5. However, in our experiment, ginsenoside Rb1 took less time to generate Rg5 compared to Rg3, and the yield of Rg5 was also higher. This indicated that the main pathway from Rb1 to Rg5 in methanol is not Rb1 → Rg3 → Rg5, but Rb1 → Rg5 directly.

Based on the above facts, we assert that we have obtained a rational overview of the conversion path that ginsenoside Rg5 is obtained through the elimination reaction of Rb1 and S-Rg3. The main route from Rb1 to Rg5 is that Rb1 directly generates Rg5, namely Rb1 → Rg5, the second route is that Rb1 first generates S-Rg3 and then generates Rg5, namely Rb1 → S-Rg3 → Rg5. The possible pathway of ginsenoside Rb1 to ginsenoside Rg5 is illustrated in Figure 4.

### 3.2. Theoretical Analysis of the Conversion Paths of Rg5

#### 3.2.1. Density Functional Theory (DFT) Calculations

Ginsenoside Rb1, S-Rg3 and R-Rg3 showed significant differences in the elimination reaction activity of producing Rg5 under acid catalysis, and surprisingly, the activity of Rb1 was stronger than S-Rg3, and S-Rg3 was stronger than R-Rg3. In order to determine the reason, the DFT study was carried out on Rb1, S-Rg3 and R-Rg3′s elimination reaction. We set the glycosyl group connected to 3C of ginsenoside Rb1, S-Rg3 and R-Rg3 as the methoxy group and analyzed the free energetics at M06-2X/def2 SV (P)/SMD (methanol) level. The results revealed that among the three relatively stable staggered conformations 1–3 of ginsenoside Rb1, S-Rg3 and R-Rg3, conformation 2, which could form a hydrogen bond between 12C-OH and 20C-OH, was the most stable optimal conformation (see Appendix A). Therefore, the free energy of E1 elimination ginsenoside Rb1, S-Rg3 and R-Rg3 in methanol was calculated at their optimal conformation 2.

The free energy spectrum of E1 elimination of ginsenoside Rb1, S-Rg3 and R-Rg3 at the optimal conformation is shown in Figure 5. Whether Rb1 or Rg3 is used as raw materials, the generation of Rg5 needs three steps. The first step is that the oxygen atom connecting 20C is protonated to protonated ether or protonated alcohol, the second step is the formation of intermediate carbocation, and the third step is the formation of Rg5 by eliminating *β*-H. Among them, the second step, which is the slowest step, is the rate determining step, so the free energy forming intermediate carbocation is mainly analyzed. From Figure 5, we showed that Gibb’s free energy for generating carbocation from protonated ginsenoside Rb1 (protonated ether) in methanol solution was 1.81kcal/mol, which was significantly lower than 7.57 kcal/mol and 7.56 kcal/mol of S-Rg3 and R-Rg3. Obviously, it is easier for ginsenoside Rb1 to generate carbocation than S-Rg3 and R-Rg3; therefore, it is easier for ginsenoside Rb1 to generate Rg5. This indicates that the DFT analysis result is consistent with the previous experimental results. This further proves theoretically that ginsenoside Rb1 can generate Rg5 more quickly and easily than Rg3, and also proves that the main generation pathway of Rg5 is Rb1 → Rg5, not Rb1 → S-Rg3 → Rg5.

Although DFT analysis can explain that Rb1 more easily generates Rg5 through an elimination reaction than Rg3, it cannot explain the experimental fact that S-Rg3 is easier to dehydrate than R-Rg3.

#### 3.2.2. Structural Analysis by Nuclear Magnetic Resonance (NMR)

Chemical reactions are influenced by many factors, such as the structure of the substrate, solvent, catalyst, temperature and reaction time. The structure of the substrates, especially the spatial structure, plays an important role in some reactions that cannot be ignored. In order to clarify the significant difference between R-Rg3 and S-Rg3 in dehydration to generate Rg5, the spatial structure of Rg3 epimers was analyzed by ^1^H-NMR, ^13^C-NMR, ^1^H-^1^H COSY, ^1^H-^13^C HSQC, ^1^H-^13^C HMBC (see Appendix A). The NMR signal dates of R-Rg3 and S-Rg3 are shown in Table 1.

The structure of ginsenoside Rg3 can be divided into three parts. The first part is the sugar moiety connected at 3C, the second part is the fused tetracyclic nucleus with rigid structure, and the third part is the aliphatic chain connected at 17C. It can be seen from Table 1 that the chemical shifts of carbon and hydrogen from 1′ to 6′′ and chemical shifts of hydrogen from 3′C–OH to 6′′C–OH in the R-Rg3 and S-Rg3 are very similar. This means that the stereo structure of R-Rg3 and S-Rg3 at the junction of the first part and the second part is similar. However, there is a significant difference between the chemical shifts of some carbon and hydrogen in the second and third parts of R-Rg3 and S-Rg3, especially the carbon and hydrogens around 20C* (see Table 2 and Table 3). For example, the chemical shifts of 22C (third part) appeared 43.35 ppm and 35.96 ppm, 21C (third part) appeared 22.80 ppm and 27.15 ppm, and 17C (second part) appeared 50.69 ppm and 54.88 ppm, respectively, in the R-Rg3 and S-Rg3. The apparent chemical shift differences (∆*δ*, ppm) between them are 7.39 ppm, −4.35 ppm and −4.19 ppm. In addition, the chemical shifts of 22H, 23H, 20C–OH and 12C–OH of R-Rg3 and S-Rg3 are also obviously different. For example, the chemical shifts of the two hydrogen atoms on the methylene group at position 22 in R-Rg3 simultaneously appears at 1.75 ppm, but it appears at 1.71 ppm (m, 1H) and 2.05 ppm (m, 1H) in S-Rg3, respectively. The chemical shifts of H23 in R-Rg3 are 2.49 ppm (m, 1H) and 2.57 ppm (m, 1H), but those in S-Rg3 are 2.30 ppm (m, 1H) and 2.62 ppm (m, 1H). This indicates that besides the differences in the configuration of 20C, S-Rg3 and R-Rg3 may also have differences in their stereo structure at the junction of the second part and the third part.

The discrepancy observed between the DFT analysis presented in Figure 5 and the differential propensity for dehydration between S-Rg3 and R-Rg3 to generate Rg5, suggests that the stable structures of R-Rg3 and S-Rg3 derived from the DFT analysis may lack accuracy. Therefore, further analysis was conducted to examine the formation of hydrogen bonds between 12C–OH (second part) and 20C–OH (third part) of R-Rg3 and S-Rg3 using NMR analysis.

In ^1^H-NMR, the hydrogen bonding effect impacts the chemical shift of ^1^H. In most cases, hydrogen bonding causes the chemical shift of ^1^H to move towards a lower field. As shown in Table 3, it can be seen that the chemical shift changes of 20C–OH and 12C–OH between R-Rg3 and S-Rg3 were 0.12 ppm and 0.07 ppm, respectively. This suggests that R-Rg3 has the potential to establish intramolecular hydrogen bonds involving the 20C–OH and 12C–OH groups, whereas S-Rg3 may lack the ability to form such intramolecular hydrogen bonds.

NOESY spectra are often used to analyze the spatial structure of organic compound molecules, such as configuration and conformation [32,33,34,35]. The basic principle is that when two nuclei are very close in space, if a double resonance experimental magnetic field (B2) is irradiated on one of the nuclei to saturate, the nuclear Overhauser effect (NOE) will occur, leading to an enhancement of the resonance signal of another nucleis [32]. If there are two hydrogen nuclei, NOE can be observed when the space distance between two hydrogen nuclei is less than 5 Å [32]. It is obvious that NOESY can provide important information for the identification of the stereo structure of organic compounds. In order to further clarify the generation of intramolecular hydrogen bonds between 20C–OH and 12C–OH, the 2D-NOESY spectrum of R-Rg3 and S-Rg3 was measured on a Bruker 600 MHz spectrometer (Appendix A). 13H (*β*-H) and 17H (*α*-H) are the hydrogen atom closest to the aliphatic chain (third part) of R-Rg3 and S-Rg3. In theory, if we understand the spatial relationship between 13H, 17H and 21H, 22H, we can clarify the spatial structure of R-Rg3 and S-Rg3. The NOE correlation of 13H and 17H, and NOE analysis of 13H, 17H with aliphatic chain’s 21H, 22H in R-Rg3 are shown in Figure 6.

From Figure 6, it can be seen that there was a strong correlation between 13H (2.03 ppm) and 21H (1.42 ppm) in R-Rg3 and a weak correlation between 17H and 22H (1.75 ppm). This indicates that the C20–C21 bond and 13H are in the same direction, which is conducive to the formation of a seven-membered intramolecularly H-bonded pseudocycle between 20C–OH and 12C–OH. This further proves the existence of intramolecular hydrogen bonds in the structure of R-Rg3. The 2D-NOESY analysis of 13H, 17H with aliphatic chain’s 21H, 22H in S-Rg3 is shown in Figure 7.

As shown in Figure 7, there was no correlation between 13H and 21H in S-Rg3, indicating that 13H and C20–C21 bonds are not in the same direction. Based on the DFT analysis results, the C20–C22 bond and 13H should adopt a codirectional structure that facilitates the formation of intramolecular hydrogen bonds, and there should also be a strong correlation between 13H and 22H (see Appendix A, S-Rg3_Conformation2).

As shown in Appendix A, there was no correlation between 13H and 22H (1.71 ppm), while the chemical shift of 13H was very close to that of another 22H (2.05 ppm), with overlapping absorption peaks, making it difficult to determine the correlation between the two hydrogen. Therefore, the correlation between 17H and surrounding H was analyzed to identify the spatial structure of S-Rg3.

As shown in Figure 7, there was a moderate correlation between 17H and 21H, while there were weak correlations between 17H and 16*α*-H or 22H. If there were intramolecular hydrogen bonds in S-Rg3, the distance between 17H and 16*α*-H was shorter than the distance between 17H and 21H (see Appendix A), which contradicts the higher correlation between 17H and 21H compared to the correlation between 17H and 16α-H. Based on the simultaneous correlation between 17H and 21H or 22H obtained from the 2D-NOESY analysis of S-Rg3, it could be inferred that the 17C–H bond formed an ortho cross structure with the C20–C21 and C20–C22 bonds, which indicates that the 20C–OH bond also forms an ortho cross structure with the C13–C17 and C16–C17 bonds (see Figure 8E). This structure causes the distance between 20–OH and 12–OH to be far apart, thus preventing the formation of intramolecular hydrogen bonds. Namely, R-Rg3 can form intramolecular hydrogen bonds, but S-Rg3, as its epimer, is not easy to form intramolecular hydrogen bonds. This means that the ability of R-Rg3 to form hydrogen bonds with methanol is reduced due to intramolecular hydrogen bonding, resulting in a significantly lower solubility of R-Rg3 in methanol solvent than S-Rg3 (see Appendix A), which significantly affects dehydration reaction activity. On the other hand, R-Rg3 also requires additional energy to disrupt intramolecular hydrogen bonds to generate Rg5, compared with S-Rg3. Obviously, the stereostructural analysis results of Rg3 epimers are consistent with the experimental results that R-Rg3 is more difficult to generate Rg5 than S-Rg3.

Why do one of the epimers of Rg3 form an intramolecular hydrogen bond and the other one does not? As shown in Figure 8A,B, if R-Rg3 forms an intramolecular hydrogen bond, it is found that the aliphatic chain (C22–C27) with the largest volume forms a Gauche conformation with C16–C17 bond, while it forms an anti conformation with C13–C17 bond, which is relatively stable (low energy). On the contrary, if S-Rg3 forms intramolecular hydrogen bonds (see Figure 8 C,D), it is found that the aliphatic chains (C22–C27) form two Gauche conformations with C16–C17 and C13–C17 bonds. Due to the significant van der waals repulsive force in the Gauche conformations, adopting two Gauche conformations is more energetically unstable than adopting one Gauche configuration and one anti conformation. That is to say, the formation of intramolecular hydrogen bonds between 20C–OH and 12C–OH is beneficial for the stability of R-Rg3 but not conducive to the stability of S-Rg3. Based on this, the dominant conformation of S-Rg3 can be inferred, as shown in Figure 8E. In the structure, the aliphatic chain (C22–C27) forms a Gauche conformation with the C16–C17 bond, while it forms an anti conformation with the C13–C17 bond, which is relatively stable in energy.

## 4. Materials and Methods

### 4.1. Materials and Instruments

Ginsenoside Rb1, S-Rg3 and R-Rg3, all at 98% purity, were purchased from the Shanghai Yuanye Biotechnology Co., Ltd. (Shanghai, China). The PPD saponins (containing 47.6% of Rb1, 19.8% of Rc, 3.1% of Rb2, containing 4.3% of Rb3 and 18.9% of Rd) were purified from a laboratory sample with an initial concentration of 80% to achieve a final concentration of 93.7%. HPLC-grade acetonitrile and methanol were purchased from the Tedia Company (Fairfield, IH, USA). Concentrated hydrochloric acid was purchased from the Tianjin Damao Chemical Reagent Factory (Tianjin, China). Purified water was purchased from Hangzhou Wahaha Group Co., Ltd. (Hangzhou, China). A silica gel 60 F254 thin-layer plate was purchased from Merck (Jena, Germany).

HPLC analysis was performed on a Dalian Elite P230 liquid chromatograph with a P230P high-pressure constant flow pump, UV230+ultraviolet-visible detector and EC-2000 LU workstation. A silica gel 60 F254 thin-layer plate (Merck) was used for thin layer chromatography (TLC). A rotary evaporator (Re-2000A, Shanghai Yarong, Shanghai, China) was used for concentration. The ^1^H-NMR, ^13^C-NMR and 2D-NMR (^1^H-^1^H COSY, ^1^H-^1^H NOESY, ^1^H-^13^C HSQC and ^1^H-^13^C HMBC) spectra were obtained on a Bruker 600 MHz instrument and the specific parameters for analysis are shown in supporting information (see Appendix A).

### 4.2. Preparation Method for Ginsenoside Rg5

In the 500 µL ginsenoside S-Rg3 (1.27 mmol/L in methanol), concentrated hydrochloric acid was added until the acid concentration of the solution was 40 mmol/L, and the mixed solution was put into a 60 °C water bath for reaction at 300 rpm for 1 h. After reacting, place it for cooling and neutralize it to pH = 6~7 with saturated sodium carbonate aqueous solution, extract three times with equal volume water saturated *n*-butanol (directly concentrate when methanol used as solvent), and then rotary evaporate it till dry under 45 °C. The remaining was dissolved in chromatographic pure methanol, and HPLC was analyzed after filtering with 0.45 μm filter head.

R-Rg3, Rb1 and PPD-type saponin also reacted and analyzed according to the above method. The amplification experiment of PPD-type saponin was conducted at 200 times the amount of the above experiment. Conversion rate and yield were carried out using the following formulas:$$\mathrm{Conversion}\;\mathrm{rate}\;(\%)=\frac{\mathrm{mass}\;\mathrm{of}\;\mathrm{raw}\;\mathrm{material}\;-\mathrm{mass}\;\mathrm{of}\;\mathrm{unreacted}\;\mathrm{raw}\;\mathrm{material}\;}{\mathrm{mass}\;\mathrm{of}\;\mathrm{raw}\;\mathrm{material}\;}\times 100$$
$$\mathrm{Yield}\;(\%)=\frac{\mathrm{actual}\;\mathrm{yield}}{\mathrm{theoretical}\;\mathrm{yield}}\times 100$$

When the raw material was the PPD-type saponin mixture, the theoretical yield of the single saponins contained in PPD saponins were first calculated separately and then added together as the total theoretical yield.

### 4.3. TLC Analyzing Method

Reaction products were monitored by TLC analysis. Solvent system: CHCl_3_/MeOH/H_2_O (65:35:10, *v*/*v*, lower phase) and spots in the silica gel plates were colored by heating at 105 °C after being sprayed with 10% sulfuric aqueous solution.

### 4.4. HPLC Analyzing Method

The chromatographic conditions were as follows [30]: Chromatography column; RESTEK Pinnacle II C18 column (5 μm, 250 × 4.6 mm); UV detector, detection wavelength 203 nm; injection volume, 20 µL; column temperature, 35 °C; volume flow, 1 mL/min; mobile phase: (A) acetonitrile, (B) water; gradient elution program: 0–10 min, 22% A; 10–20 min, 22–27% A; 20–25 min, 27–31% A; 25–45 min, 31–38% A; 45–60 min, 38–52% A; 60–65 min, 52% A; 65–75 min, 52–55% A; 75–82 min, 55–60% A; 82–82.10 min, 60–90% A; 82.10–100 min, 90% A; 100–100.10 min, 90–22% A; 100.10–115 min, 22% A.

### 4.5. Density Functional Theory (DFT) Analyzing Method

All calculations were carried out with Gaussian 09 software [36]. The M06-2X function [37,38] was adopted for all calculations. For geometry optimization and frequency calculations, the def2-SVP basis set [39] was used, and the optimal geometry for each compound was determined. Singlet point energy calculations were performed with a larger basis set def2-TZVP basis set [39]. The SMD implicit solvation model [40] was used to account for the solvation effect of methanol.

### 4.6. Amplification Experiment for Preparation of Ginsenoside Rg5

In the 100 mL PPD saponins solutions (1.5 mg/mL in methanol), concentrated hydrochloric acid was added until the acid concentration was 40 mmol/L, and the mixed solution was placed into a 60 °C water bath for reaction at 300 rpm for 20 min. After reacting, the reaction product was used to carry out post-processing according to the method in Section 4.2, and HPLC analysis was performed.

## 5. Conclusions

Ginsenoside Rg5 is a very important secondary ginsenoside with good pharmacological activity. Due to a lack of in-depth understanding of the conversion pathway of ginsenoside Rg5, the process of preparing ginsenoside Rg5 using different raw materials has always been accompanied by low yield and high cost. According to our experimental results, ginsenoside Rb1 can be rapidly converted into Rg5 in alcohol solvent, followed by S-Rg3, while R-Rg3 is very difficult. The main pathway from Rb1 to Rg5 in methanol by hydrochloric acid catalyst is not Rb1 → S-Rg3 → Rg5, but Rb1 → Rg5 directly. The reason why Rb1 is easier to generate Rg5 than Rg3 is that the free energy of Rb1 to generate carbocation is far lower than that of Rg3. The main reason why S-Rg3 is easier to generate Rg5 than R-Rg3 is that S-Rg3 does not form intramolecular hydrogen bonds between 20C–OH and 12C–OH, so it does not need to provide additional energy, like R-Rg3, to destroy intramolecular hydrogen bonds, so the energy required to generate carbocation is lower than R-Rg3. Another reason is that S-Rg3 has a higher solubility than R-Rg3 in the reaction solvent. The magnification experiment results using PPD saponins as raw materials show that their reaction effect in generating Rg5 is similar to that of Rb1. The experiment shows that PPD saponins can complete the transformation to Rg5 within 20 min, and the yield remains at a high level. Obviously, the utilization of PPD saponins for the preparation of ginsenoside Rg5 not only offers significant time savings and operational simplification but also demonstrates a notable enhancement in the yield of Rg5 when compared to Rg3. Clarifying the pathway and theoretical research of ginsenoside conversion to Rg5 are of great significance for the efficient and low-cost preparation of the bioactive substance ginsenoside Rg5, which will undoubtedly promote extensive research on the pharmacological activity of ginsenoside Rg5 and the development of related drugs.

## Data Availability

Not applicable.

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
