# Peer review of "How to More Effectively Obtain Ginsenoside Rg5: Understanding Pathways of Conversion"

_molecules, 2023, doi:10.3390/molecules28217313_

Round 1

Reviewer 1 Report

Comments and Suggestions for Authors

In this review, I evaluate a manuscript titled "How to more effectively obtain Ginsenoside Rg5: Understanding Pathways of Conversion" The manuscript presents a comprehensive investigation into the conversion pathways of ginsenosides and the factors influencing their transformation into ginsenoside Rg5. While the study is intriguing and has the potential to contribute to our understanding of ginsenoside chemistry, there are several critical issues that need to be addressed before it can be considered for publication.

Clarity and Organization: The manuscript lacks clarity and proper organization. It is crucial to restructure the manuscript with a clear introduction, methodology section, results, discussion, and conclusion. The current organization makes it challenging for the reader to follow the logical flow of the research.

Figures and Tables: The figures and tables included in the manuscript are of utmost importance in conveying the research findings. However, several issues need to be addressed:

Figure and Table Legends: The legends for figures and tables need to be more descriptive. They should explain what each figure or table illustrates without requiring the reader to reference the text constantly.

Methodology and Data Presentation: The methodology section should be expanded and clarified to provide more details on the experimental procedures, such as the exact conditions and equipment used for the reactions. In scientific research, it's essential to document the parameters of your experiments thoroughly to ensure reproducibility and accurate data interpretation. Here's a brief overview of the key parameters for the mentioned NMR and 2D NMR experiments:

1H-NMR (Proton Nuclear Magnetic Resonance)

Spectrometer frequency (MHz)

Solvent used

Sample concentration (mg/mL)

Temperature (°C)

Number of scans

Pulse width (π/2 pulse)

Relaxation delay (D1)

Acquisition time

Data processing and phase correction details

13C NMR (Carbon-13 Nuclear Magnetic Resonance)

Spectrometer frequency (MHz)

Solvent used

Sample concentration (mg/mL)

Temperature (°C)

Number of scans

Pulse width (π pulse)

Relaxation delay (D1)

Acquisition time

Data processing and phase correction details

1H-1H COSY (COrrelation SpectroscopY)

Same parameters as 1H-NMR for each spectrum involved in the COSY experiment (e.g., the two 1H-NMR spectra that are being correlated).

1H-1H NOESY (Nuclear Overhauser Effect SpectroscopY)

Similar parameters to 1H-NMR, with the addition of mixing time for NOESY (ms)

Data processing and phase correction details

1H-13C HSQC (Heteronuclear Single Quantum Coherence)

Spectrometer frequency for both 1H and 13C nuclei

Solvent used

Sample concentration (mg/mL)

Temperature (°C)

Number of scans

Pulse widths for both 1H and 13C

Relaxation delay (D1)

Acquisition time

Data processing and phase correction details

1H-13C HMBC (Heteronuclear Multiple Bond Correlation)

Similar parameters to 1H-13C HSQC, with the addition of the long-range coupling delay (J-coupling delay)

Data processing and phase correction details

Detailed documentation of these parameters is crucial for ensuring the accuracy and reliability of your NMR experiments. It also helps other researchers understand and reproduce your work if necessary.

Discussion and Implications: The manuscript should include a more in-depth discussion of the implications of the findings. How do the results contribute to the field of ginsenoside research? What are the practical applications or significance of the observed differences in transformation pathways between 20S-Rg3 and 20R-Rg3?

The current exposition of the concepts behind the NOESY experiment is really concise. "In NMR analysis, when the space distance between Hx and Hy atoms in organic compounds is less than 5Å , NOE will occur, that is, when irradiating Hx, the intensity of Hy will increase. So, NOESY can provide important information for the identification of the stereo structure of organic compounds." I would recommend perusing the most recent scholarly publications pertaining to this subject matter.[10.1016/j.molliq.2023.122620 ] [10.1007/s00723-023-01608-w] [10.1016/j.molliq.2023.122230]

Clarity of Scientific Language: The manuscript would benefit from using more precise scientific language. For instance, the phrase "higher than 10.59%" should be replaced with the exact percentage to enhance clarity.

In summary, while the manuscript addresses an interesting topic in ginsenoside research, it requires substantial revisions to enhance clarity, organization, and presentation. Once these issues are addressed, the manuscript has the potential to make a valuable contribution to the field. I recommend a major revision with attention to the points mentioned above. After the necessary revisions, the manuscript can be re-submitted for further evaluation.

Reviewer 2 Report

Comments and Suggestions for Authors

Review of the paper molecules-2615537 entitled How to more effectively obtain Ginsenoside Rg5: Understanding Pathways of Conversioncompared different conversion pathways using Rb1, R-Rg3 and S-Rg3 as the raw material to obtain Rg5 under simple acid catalysis. The experimental data and theoretical calculations of free energies suggested that the reaction activity of the Rg5 conversion following the order of Rb1>S-Rg3>R-Rg3. Thus, protopanaxadiol (PPD)-type ginsenosides could be used as synthetic precursors for Rg5. Even the topic is interesting, the reviewer has some questions / comments concerning this work:

-      The authors tried to use simple acid catalysis for the ginsenosides conversion, which in industrial the biotransformation by some microorganisms are normally being used, the conversion free energies might not fit for the biotransformation?

-The literature “Foods 202312(12), 2349” “ Frontiers in Nutrition, 2022, 9: 833859.” reported amino acids acted as catalysts for the conversion of PPD type saponins to Rg5 the yield could reach a maximum of around 10% , which sounds reasonable. In this study, the results showed that the yield of Rg5 reached 47.71% under simple acid catalysis, HPLC integration of the standards and conversion products should be carried out at the same sample concentration.

-The structure of Rg1 and conversion pathway should be illustrated in figures.

- Fig. 3 should be adjusted to Fig. 2C.

Round 2

Reviewer 1 Report

Comments and Suggestions for Authors

The authors have effectively addressed and clarified several aspects that were previously unclear in the earlier version of the manuscript. Their response demonstrates a commendable effort to enhance the quality and comprehensibility of the research.